# Carbohydrate Mouth Rinse Mitigates Mental Fatigue Effects on Maximal Incremental Test Performance, but Not in Cortical Alterations

**DOI:** 10.3390/brainsci10080493

**Published:** 2020-07-29

**Authors:** Cayque Brietzke, Paulo Estevão Franco-Alvarenga, Raul Canestri, Márcio Fagundes Goethel, Ítalo Vínicius, Vitor de Salles Painelli, Tony Meireles Santos, Florentina Johanna Hettinga, Flávio Oliveira Pires

**Affiliations:** 1Exercise Psychophysiology Research Group, School of Arts, Sciences and Humanities, University of São Paulo, São Paulo 05508-060, Brazil; cayquebbarreto@alumni.usp.br (C.B.); francope@gmail.com (P.E.F.-A.); raulcanestri@usp.br (R.C.); gbiomech@usp.br (M.F.G.); italovinicius@usp.br (Í.V.); vitor_pa@hotmail.com (V.d.S.P.); 2Human Movement Science and Rehabilitation Program, Federal University of São Paulo, Avenida Sena Madureira 1500, Brazil; 3Porto Biomechanics Laboratory (LABIOMEP), University of Porto, 4000 Porto, Portugal; 4Strength Training Study and Research Group, Institute of Health Sciences, Paulista University, Sao Paulo 05347-020, Brazil; 5Physical Education Program, Research Center for Performance and Health, Federal University of Pernambuco, Recife 52071-030, Brazil; tonymsantos@gmail.com; 6Department of Sport, Exercise and Rehabilitation, Northumbria University, Newcastle NE1 8ST, UK; florentina.hettinga@northumbria.ac.uk

**Keywords:** brain regulation, physical performance, cognitive performance, supplementation

## Abstract

Detrimental mental fatigue effects on exercise performance have been documented in constant workload and time trial exercises, but effects on a maximal incremental test (MIT) remain poorly investigated. Mental fatigue-reduced exercise performance is related to an increased effort sensation, likely due to a reduced prefrontal cortex (PFC) activation and inhibited spontaneous behavior. Interestingly, only a few studies verified if centrally active compounds may mitigate such effects. For example, carbohydrate (CHO) mouth rinse potentiates exercise performance and reduces effort sensation, likely through its effects on PFC activation. However, it is unknown if this centrally mediated effect of CHO mouth rinse may mitigate mental fatigue-reduced exercise performance. After a proof-of-principle study, showing a mental fatigue-reduced MIT performance, we observed that CHO mouth rinse mitigated MIT performance reductions in mentally fatigued cyclists, regardless of PFC alterations. When compared to placebo, mentally fatigued cyclists improved MIT performance by 2.24–2.33% when rinsing their mouth with CHO during MIT. However, PFC and motor cortex activation during MIT in both CHO and placebo mouth rinses were greater than in mental fatigue. Results showed that CHO mouth rinse mitigated the mental fatigue-reduced MIT performance, but challenged the role of CHO mouth rinse on PFC and motor cortex activation.

## 1. Introduction

Mental fatigue is a mental state caused by a prolonged, highly demanding cognitive task [1] that induces an increased fatigue sensation and reduced focus on a given task [1,2,3]. From a physical exercise perspective, mental fatigue has been associated with a reduced exercise capacity [1,2,3] likely through its effects on cerebral activation [2,3,4] and effort sensation [2,5]. A recent systematic review has confirmed that detrimental effects of mental fatigue on exercise performance have been well evidenced in endurance exercise modes, such as constant workload and time trial exercises [1,3,5]; however, the effects on maximal incremental test (MIT) performance remain poorly investigated [6,7]. This is a relevant aspect, as MIT is a gold-standard protocol, with intensity gradually increased until exhaustion, to assess cardiopulmonary fitness variables, such as maximal oxygen uptake (VO_2MAX_) and ventilatory thresholds [8]. Accordingly, potential deleterious mental fatigue effects on MIT performance and cardiopulmonary fitness variables may be important in clinical and sport settings [8,9] and require more investigation.

Despite the number of studies confirming the detrimental effects of mental fatigue on endurance exercise performance [1], only a few have investigated if centrally active compounds may mitigate such effects [2,10]. The suggestion that centrally active compounds may counteract mental fatigue effects is based on the fact that mental fatigue is associated with a reduced prefrontal cortex (PFC) activation [2,3,4] and inhibits spontaneous behavior [11] due to the increased cerebral ATP hydrolysis and adenosine concentration [12,13]. In a physical exercise scenario, centrally active compounds capable of attenuating the effects of detrimental mental fatigue on cerebral activation may thus be of interest. Recently, Franco-Alvarenga et al. [2] found that caffeine ingestion mitigated the mental fatigue-derived performance reduction in a 20 km cycling time trial. Interestingly, they also found that caffeine-attenuated exercise performance reduction was unrelated to PFC activation, an area involved in perceptual [14], attentional, and inhibitory responses [15]. These results challenged the underlying mechanisms of caffeine effects on exercise performance in mentally fatigued individuals, thus highlighting the need for more investigations on this topic.

Since a seminal study by Carter et al. in 2004 [16], a number of studies have confirmed that carbohydrate (CHO) mouth rinse may potentiate endurance exercise performance [17] likely due to its centrally mediated effects. The underlying mechanisms of the central effects of CHO mouth rinse involve the activation of cerebral areas related to motor planning and emotional responses, such as PFC [18,19]. Therefore, one may suggest that CHO mouth rinse may potentially mitigate the deleterious mental fatigue effects on PFC activation and exercise performance through its beneficial effects on cerebral responses [18,20], endurance exercise performance, and effort sensation [17]. Indeed, a recent study provided insightful results of the CHO mouth rinse effects on cerebral, cognitive, and perceptual responses, as participants improved the accuracy of answers and lowered the mental fatigue sensation when they performed a highly demanding cognitive task while regularly rinsing their mouth with a caffeine-combined CHO solution [20]. Interestingly, the mouth rinses improved some cerebral responses in PFC and other cortex areas, although unchanged responses were observed in other cortex areas. Therefore, despite using a caffeine-combined CHO solution, rather than a CHO solution in isolation, these results may shed light on the potentially beneficial CHO mouth rinse effects on cerebral and perceptual responses in mentally fatigued individuals. However, that study used no exercise performance, so the potential of CHO mouth rinse to mitigate deleterious mental fatigue effects on exercise performance is still unknown.

Two important aspects to consider when designing a straightforward methodology to unravel the CHO mouth rinse effects on exercise performance in mentally fatigued individuals are the use of electroencephalography (EEG) measures and controlled exercise mode. Firstly, it has been observed that an increased EEG theta wave over the PFC is a sensitive method to identify a mental fatigue state at rest [3,4], as this area is involved in inhibitory and sustained attention responses [21]. Moreover, analysis of the entire EEG spectrum may also be insightful in an exercise scenario, as PFC activation has been associated with multiple functions, such as exercise-induced perceptual responses [14], attentional and inhibitory control [15,22], and motor cortex (MC) activation [3,23]. However, given that both CHO mouth rinse and mental fatigue have been suggested to change PFC activation [3,18], it remains to be investigated if rinsing the mouth with CHO may counteract the mental fatigue effects on PFC and MC activation and exercise performance.

Secondly, the use of MIT as exercise mode may be methodologically sound to investigate the mental fatigue-CHO mouth rinse interplay in endurance exercise performance. Different from a ramp test, a graded MIT provides a number of controlled, constant workloads at increasing intensities, thereby allowing studying the CHO mouth rinse effects on physiological variables in a number of controlled intensities. This aspect is relevant, as a study showed that CHO mouth rinse potentiated performance in moderate rather than severe intensities [24], thus indicating the necessity to study the effects of CHO mouth rinse at different exercise intensities. Furthermore, an MIT would allow knowing how the mental fatigue-CHO mouth rinse interplay affects peak power output (W_PEAK_), peak oxygen comsumption (VO_2PEAK_), and ventilatory thresholds, a topic poorly investigated, as mentioned earlier.

Therefore, the present study aimed to verify if rinsing the mouth with CHO solution may counteract the deleterious mental fatigue effects on MIT performance and PFC and MC activation. We also aimed to examine if CHO mouth rinse in mentally fatigued individuals may change MIT outcomes, such as W_PEAK_, VO_2PEAK_, and ventilatory thresholds, such as the first (VT_1_) and second ventilatory threshold (VT_2_). We hypothesized that mental fatigue would negatively affect MIT outcomes; however, CHO mouth rinse would mitigate these effects.

## 2. Materials and Methods

### 2.1. Participants and Ethics

The sample size was calculated through equation suggested elsewhere as *n* = 8*e*^2^/*d*^2^ [25], where *n*, *e*, and *d* represent sample size, coefficient of variation, and percentage of treatment magnitude, respectively. Participants were invited through social media. After identification of eligible participants, 20 well-trained male cyclists, who attained a W_PEAK_ greater than 325 W in a preliminary MIT, took part in this study. Cyclists were classified within performance levels 2 and 3, according to classification suggested elsewhere [26], and were engaged in regional and national level competitions (7.56 yr ± 5.89 of cycling experience and 310.6 km·week^−1^ ± 128.1 of training volume) at the time the study was conducted. Cyclists should be free from cardiopulmonary, metabolic, and orthopedic diseases and being non-user of prohibited substances. All participants were informed about the risks and benefits before signing the informed written consent. This study was approved by the local Ethics Committee (54910716.4.0000.5390) and conformed the Declaration of Helsinki.

### 2.2. Study Design

This crossover, counterbalanced, CHO-perceived placebo-controlled study submitted cyclists to 3 experimental sessions after a preliminary and a baseline MIT session (Figure 1). In the first session, cyclists completed an MIT until exhaustion to identify eligible cyclists and habituate them with the MIT procedures. We selected cyclists attaining a W_PEAK_ ≥ 325 W in this preliminary MIT session, as we intended to reduce the between-subjects performance variability. Additionally, this preliminary MIT served to calculate the percentage of W_PEAK_ (%W_PEAK_) at which the participants rinsed their mouth with CHO or PLA in the experimental MIT. In the second session, cyclists performed a baseline MIT (BASE), while in sessions 3–5, they performed an MIT with mental fatigue (MF), or combining MF+CHO and MF+placebo (PLA). Importantly, instead of having 1 min steps throughout the MIT protocol, we used extended steps during the MIT in MF, MF+CHO, and MF+PLA sessions, so that steps at 25%, 50%, and 75% of the W_PEAK_ determined in preliminary MIT had 2 min duration. This adaptation was necessary to make possible the CHO mouth rinses every 25% of the trial while recording steady EEG signals during controlled constant workload at different intensities. Sessions 1 and 2 were performed in sequential order, but sessions 3 to 5 were performed in a counterbalanced order. Sessions 2 and 3 provided a proof-of-principle study of the mental fatigue in MIT, as this comparison allowed us to assess deleterious mental fatigue effects on MIT outcomes before studying the combined CHO mouth rinse-mental fatigue effects on this exercise mode.

Responses, such as gas exchange, as well as PFC and MC EEG, were continuously assessed throughout the MIT, while ratings of perceived exertion (RPE) were obtained at regular intervals. Additionally, EEG, fatigue sensation, and motivation were further obtained before and immediately after the cognitive test-induced mental fatigue. Cyclists were instructed to maintain their regular diet, avoid exhaustive exercise, and refrain from stimulant substances (i.e., energy drink, caffeine, pre-workout beverages) and alcoholic beverages for the 24 h before the sessions. The sessions were performed after a 6–8 h fasting period at the same time of day in a controlled temperature (~22 °C) and relative humidity (~60%) environment. A 3–7 days wash-out interval was observed between sessions.

### 2.3. Maximal Incremental Test (MIT)

All the MITs were performed on a road bike (Giant^®^, New York, NY, USA) attached to a cycle simulator (Computrainer, Racer-Mate^®^ 8000, Seattle, WA, USA), calibrated according to manufacturer′s instruction before every test. Seat and handlebar positions were individually adjusted before every test, following individual adjustment of the preliminary session. After a 5 min controlled-pace warm-up (100 W at 80 rpm), the workload was increased to 25 W∙min^−1^ during the continued MIT until exhaustion, defined as the incapacity to maintain the pedal cadence despite three verbal encouragements (consisted of standardized words). These elongated steps allowed the record of a low-noise, steady time-matched EEG signal just after the mouth rinses. Cyclists were previously familiarized to keep their eyes closed and avoid excessive upper limb movements during exercise at these intensities.

### 2.4. Mental Fatigue Protocol

In accordance with previous studies [2,3], we used the rapid visual information processing test (RVIP), a high-demanding, sustained attention, and inhibitory control task to induce a mental fatigue state. Cyclists sat down in a comfortable chair in front of a 17-inch monitor, in a quiet and illuminated environment, wearing an earplug to reduce environment noises [2]. The RVIP lasted 40 min and consisted of identifying sequential three odd or even numbers randomly showed in a frequency of one number per 600 ms. Participants used the spacebar of a standard keyboard to indicate the correct answers. Sequences of three odd (i.e., 7, 3, 1; 1, 9, 5) or even (i.e., 8, 6, 2; 2, 8, 4) numbers were shown 8 times per minute. False alarms (a.u), accuracy answers (%), and reaction time (s) were used to assess cognitive performance [2].

### 2.5. Mouth Rinsing Protocols

Following recommendations from previous studies [18,19,27], we used artificial saliva (0.21 g of NAHCO_3_ and 1.875 g of KCl per 1 L of ddH_2_O) as a base for both CHO and PLA solutions [19,28]. While PLA was formulated as 0.053 g of acesulfame K diluted in 1 L of artificial saliva, the CHO solution was formulated, having 64 g of maltodextrin diluted in 1 L of PLA solution. Before the study, researchers tested these solutions to test the blinding efficacy. The cyclists were asked to rinse their mouth for 10 s with 25 mL CHO or PLA solution at the end of the standard warm-up as well as at 25%, 50%, and 75% of the MIT. The volume was spat into a bowl after each mouth rinse, and then we checked the eventual ingestion of substances by measuring the volume through a 10 mL syringe.

Importantly, we designed this study having a CHO-perceived placebo and a baseline condition as controls, in accordance with a recent placebo intervention consensus [29], suggesting that the expectation of receiving an active compound may be a bias source in sports nutrition studies [30]. Earlier studies had challenged the traditional double-blind, clinical trial design to investigate the ergogenic aid effects on physical performance [31,32], given that the expectancy itself could result in physical performance improvements [33]. In contrast, it has been suggested that the use of an active substance-perceived placebo condition may work as a control for the expectation-induced performance alterations in ergogenic supplement studies [29,30]. Therefore, in the present study, we reinforced the suggestion that CHO mouth rinse is a potential ergogenic aid to improve performance, leading cyclists to believe they would wash their mouths with CHO solution in all experimental sessions (i.e., CHO and PLA trials). We also included a baseline session (BASE) in the design, allowing us to know the effect size of both CHO and PLA mouth rinses relative to a condition totally inert, which had neither expectations nor pharmacological effects.

### 2.6. Measures and Instruments

#### 2.6.1. Physical Performance

The performance was indicated by W_PEAK_ and time to exhaustion. The W_PEAK_ was defined as the highest power output attained in the last completed stage (60 s), corrected by the time (s) spent in an incomplete stage when necessary. The time to exhaustion was determined when cyclists could no longer maintain the target pedal cadence, despite three strong verbal encouragements provided by a researcher unaware of the substance used in the mouth rinses. Instructions and words used to verbally encourage cyclists were previously standardized.

#### 2.6.2. Gaseous Exchange

The gaseous exchange was recorded breath-by-breath through a mask (Hans Rudolph, Shawnee, KS, USA) coupled to an open circuit gas analyzer (Metalyzer 3B, Cortex, Leipzig, Germany) for minute ventilation (VE), oxygen uptake (VO_2_), and carbon dioxide production (VCO_2_). The expired air was measured by using a bi-directional flow sensor calibrated before every test. A zirconium sensor analyzed the expired O_2_, while the end-tidal CO_2_ was analyzed through an infrared sensor. Sensors were calibrated according to the manufacturer’s guidelines through a known O_2_ (12%) and CO_2_ (5%) concentration. Importantly, the mask was briefly removed from the participants’ faces at the extended 2 min steps (~15 s), and thus they could rinse their mouth with CHO and placebo solutions. Researchers involved in this part of the study were trained during pilot tests to ensure minimal artifacts during the gaseous exchange and EEG data sampling. Breath-by-breath data were averaged as 10 s intervals so that VO_2PEAK_ and maximal respiratory exchange ratio (RER_MAX_) were determined as the mean values over the last 30 s of the test during presumed maximal effort [34]. Two experienced researchers visually identified the VT_1_ and VT_2_ by analyzing the VE/VO_2_ and VE/VCO_2_ curves, at the first breakpoint in VE/VO_2_ and last breakpoint before a systematic raise in VE/VCO_2_, respectively. Both RER and VE breakpoints, defined as a systematic raise in these variables, were used to confirm VT_1_ and VT_2_, respectively. The median between researchers was used in cases of non-agreement between evaluators. Then, thresholds were expressed as absolute power output and VO_2_ values, as well as relative to W_PEAK_ and VO_2PEAK_.

#### 2.6.3. Electroencephalography

Active electrodes (Ag-AgCl) were placed at Fp1 and Cz positions following the EEG international 10–20 system, oriented by *nasion* and *inion* positions, and referenced to the mastoid process [35]. Electrodes were fixed with medical strips after exfoliation, cleaning, and application of the conductive gel. Resting EEG signal (Emsa®, EEG BNT 36, TiEEG, Rio de Janeiro, Brazil) was recorded during a 180 s time window before and immediately after the RVIP test completion, when cyclists were in absolute rest, with eyes closed, without body or facial movement. Importantly, they maintained their eyes closed during the extended 2 min stages, thus avoiding excessive facial and upper limb movements during EEG capture during MIT. Briefly, a reduced PFC activation, assessed specifically through an increased EEG theta band (3–7 Hz), has been associated with a highly demanding cognitive task-induced mental fatigue [2,3,36]. Hence, we used an increased activity in this particular EEG band over the PFC from pre to post RVIP test to assess mental fatigue. Moreover, PFC activation has been also suggested to play a role in exercise regulation [14,15], as PFC is involved in exercise-induced perceptual responses [11], attentional and inhibitory control [12], and MC activation during exercise [3,13,14]. Earlier studies have suggested that CHO mouth rinse has increased brain activation in PFC [18], so we hypothesized that cortical activation, as measured through the entire EEG power spectrum, would provide a reliable picture of the effects of CHO mouth rinse on cortical activation in mentally fatigued cyclists performing the exercise.

The EEG signal was captured with a Notch filter, before data filtering through a 3–50 Hz band-pass recursive filter. The EEG signal recorded before and after the RVIP test was processed within a 5 s time window (1800 samples) of the most steady signal (lowest local standard deviation) within a −200 and 200 μV amplitude range, after removing the initial and final 30 s time window, a period containing noises related to the individuals’ expectancy regarding the start and end of the EEG protocol [37]. The power spectrum within the 3–7 Hz was calculated in PFC through a Fast Fourier Transform so that an increase of the power spectrum within this particular EEG band was interpreted as evidence of mental fatigue [2,3,4]. In contrast, we further calculated the total power spectrum density (PSD) within the 3–50 Hz band in a 5 s time window in the most steady signal captured immediately after each mouth rinse at the end of the warm-up and at 25%, 50%, and 75% W_PEAK_ during MIT. The use of the entire EEG power spectrum is more meaningful to indicate the exercise-induced alteration in cortical activity. Resting EEG data were expressed as absolute values (dB), while exercise EEG data were expressed as a change (%) from the baseline values recorded before the RVIP test [23,38].

#### 2.6.4. Psychological Responses

Mental fatigue sensation was assessed through a visual analog scale (VAS) before and after the RVIP test. Cyclists were asked to answer “how mentally fatigued you feel now” by using a scale ranging from “0” to “100” mm to rate mental fatigue sensation as “none at all” and “maximal”, respectively [2,39]. Moreover, we assessed RPE at the end of extended MIT steps at 25%, 50%, and 75% W_PEAK_ through the 15-point Borg’s scale [40], having its anchors as reported elsewhere [41]. The slope of the RPE-exercise relationship (RPE_SLOPE_) was used to indicate how RPE progressed during exercise, while RPE_MAX_ ≥ 18 indicated if the maximal effort was attained.

## 3. Statistical Analysis

Data were firstly checked for Gaussian distribution; thereafter, they were presented as mean and standard deviation (SD). As a proof-of-principle of the mental fatigue effects on neurophysiological, cognitive, perceptual, and exercise performance variables, we performed a series of comparisons involving BASE and MF exercise sessions before investigating the CHO mouth rinse effects on mentally fatigued individuals. Then, using the MF exercise session, we firstly confirmed the increase of EEG theta band in PFC and mental fatigue sensation from pre to post RVIP test through a paired student *t*-test. Accordingly, we confirmed a likely impairment in cognitive performance over the RVIP test, and thus we compared reaction time, accurate answers, and false alarms through a repeated-measures mixed model design. Secondly, using BASE and MF exercise sessions, we compared MIT outcomes, such as W_PEAK_, time to exhaustion, VO_2PEAK_, and RPE_SLOPE_, through a paired student *t*-test. Given the dependence nature between thresholds, we compared VT_1_ and VT_2_ between BASE and MF through a repeated-measures mixed model design. In mixed models analysis, we used the restrict Likelihood log criteria to find the covariance matrix that best fitted to the dataset, and cases of significant F-values were corrected by Bonferroni’s test. Additionally, we used the BASE MIT session to calculate the mental fatigue-derived smallest worthwhile change on performance, as suggested elsewhere [42].

Using MF, MF+CHO, and MF+PLA exercise sessions, we verified the CHO mouth rinse effects on MIT outcomes, such as W_PEAK_, time to exhaustion, VO_2PEAK_, RPE_SLOPE_, VT_1_, and VT_2_, in mentally fatigued cyclists through a series of paired student t-tests and repeated-measures mixed models, as detailed above. We also verified the CHO mouth rinse effects on PFC and MC responses to exercise (warm-up, 25%, 50%, 75%, and 100% of the MIT) in mentally fatigued cyclists through repeated-measures mixed model comparisons.

Even performing a prior sample size estimation, we reported the post hoc effect size (ES) to confirm this initial estimation. Regardless of the statistic family test, ES was expressed as Cohens’ *d* and interpreted as small (0 ≥ *d* ≤ 0.2), moderate (0.3 ≥ *d* ≤ 0.6), large (0.7 ≥ *d* ≤ 1.2), very large (1.2 ≥ *d* ≤ 1.9), and extremely large ES (*d* ≥ 2), as suggested elsewhere [43]. Statistical significance was set at *p* ≤ 0.05, and the analysis was carried out with specific software (v.22 SPSS software, IBM, New York, NY, USA).

## 4. Results

Twenty-six eligible participants matching the W_PEAK_ criteria were picked out from a broader sample (*n* = 53), after the preliminary MIT. However, six eligible cyclists dropped out from experimental procedures due to personal reasons. Therefore, 20 cyclists (W_PEAK_ of 358.63 W ± 21.96, age of 35 yr ± 7, body mass of 80.5 kg ± 10.4, the height of 176 cm ± 5 and body mass index of 26.04 ± 2.94) completed the experimental procedures.

### 4.1. Proof-of-Principle of the Mental Fatigue Effects

When compared to pre RVIP test measures, cyclists showed an increased EEG theta band (*p* < 0.001, d = 0.47) and mental fatigue sensation (23.00 ± 24.47 vs. 42.32 ± 26.14, *p* = 0.005) after the cognitive test. Accordingly, the cognitive performance decreased throughout the RVIP test, as the time main effect revealed an impairment in reaction time (*p* = 0.04, d = 2.05) and accurate answers (*p* = 0.03, d = 0.70), although no effect was detected in false alarms (*p* = 0.60).

Comparisons involving BASE and MF sessions revealed that MIT performance was impaired in the MF session (Table 1), either expressed as time to exhaustion (*p* = 0.002, d = 0.24) or W_PEAK_ (*p* = 0.002, d = 0.25). The mean W_PEAK_ difference from BASE to MF trial was 6.99 W, being greater than the smallest worthwhile change estimated as 4.97 W. In contrast, VO_2PEAK_ responses were comparable between BASE and MF trials (*p* = 0.80). Comparisons involving thresholds showed that VT_1_ was obviously lower than VT_2_ regardless of the condition, and thus the time effects were omitted. Neither VT_1_ nor VT_2_, expressed either in absolute power output (*p* > 0.05) and VO_2_ values (*p* > 0.05) or relative to peak values (i.e., W_PEAK_, *p* > 0.05; VO_2PEAK_, *p* > 0.05), were significantly different between BASE and MF sessions. Accordingly, no time by condition interaction effect was observed in VT_1_ (*p* > 0.05 for all results) and VT_2_ (*p* > 0.05 for all results). The RPE_SLOPE_ was comparable between MF and BASE trials (*p* = 0.62). Table 1 depicts all MIT outcomes between the BASE and MF trials.

### 4.2. Mental Fatigue Responses before the MIT in Mouth Rinse Sessions

In accordance with proof-of-principle results, we also observed an increased EEG theta band in both MF+CHO (*p* = 0.02, d = 0.56) and MF+PLA sessions (*p* < 0.001, d = 0.71) from pre to post RVIP test. Likewise, mental fatigue sensation increased from pre to post RVIP test in both MF+CHO (10.82 ± 11.38 vs. 40.58 ± 25.30; *p* < 0.001, d = 1.44) and MF+PLA sessions (17.18 ± 14.23 vs. 47.65 ± 23.65; *p* < 0.001, d = 1.49). Together, cognitive performance worsened similarly throughout the RVIP test in both MF+CHO and MF+PLA mouth rinse sessions, as indicated by a time main effect in reaction time (*p* = 0.003, d = 0.76). In contrast, no time effects were observed in accurate answers (*p* = 0.12) and false alarms (*p* = 0.66). Given that no mouth rinses were performed during the cognitive test, no session main effect was observed in reaction time (*p* = 0.58), accurate answers (*p* = 0.25), or false alarms (*p* = 0.06).

### 4.3. CHO Mouth Rinse Effects on MIT Outcomes in Mentally Fatigued Cyclists

We observed a significant mouth rinse main effect in W_PEAK_ (343.85 ± 27.38 vs. 344.52 ± 24.00 vs. 350.21 ± 21.20 W for MF, PLA+MF, and CHO+MF respectively, *p* = 0.01, d = 0.72) and time to exhaustion (825.75 ± 65.62 vs. 826.79 ± 57.64 vs. 839.74 ± 50.08 s for MF, PLA+MF, and CHO+MF respectively, *p* = 0.01, d = 0.70), showing that CHO mouth rinse attenuated the mental fatigue-reduced MIT performance (Figure 2). Thus, as depicted in Figure 3, both the W_PEAK_ and time to exhaustion were greater in MF+CHO than MF+PLA (*p* = 0.04 and *p* = 0.04) and MF session (*p* = 0.02 and *p* = 0.03), although no difference was observed in VO_2PEAK_ (*p* = 0.37). Regarding the ventilatory thresholds, VT_1_ was obviously lower than VT_2_ regardless of the condition, and thus the time effects were omitted. No difference was observed between MF and MF+CHO trial for VT_1_ and VT_2_, regardless of the variable used to express them. However, VT_1_ and VT_2_ were greater in MF+PLA than MF trial when expressed as absolute or relative power output values, but not as absolute and relative VO_2_. No difference was detected between thresholds identified in MF+CHO and MF+PLA. Comparable RPE_SLOPE_ responses were observed in MF, MF+CHO, and MF+PLA trials (*p* = 0.79). Table 2 shows the MIT outcomes found in MF, MF+CHO, and MF+PLA trials, except for those outcomes earlier reported as figures.

Regarding PFC activation during exercise, we found a mouth rinse (*p* < 0.001, d = 1.16) and a time main effect in EEG responses (*p* < 0.001, d = 1.02), as both CHO and PLA mouth rinses changed the PFC activation of mentally fatigued cyclists throughout the MIT. Multiple comparisons showed that PFC activity was higher in MF+CHO (*p* = 0.01) and MF+PLA (*p* < 0.001) than MF, but no difference was detected between MF+CHO and MF+PLA (*p* = 0.95). Time by mouth rinses interaction effects revealed that PFC activity was greater in MF+CHO than MF at 75% of the MIT (*p* = 0.01). Likewise, PFC activity was greater in MF+PLA than MF at 50% (*p* = 0.02) and 75% of the MIT (*p* = 0.01). Figure 3 (upper panel) depicts the results of PFC activation during MIT.

Accordingly, we found a mouth rinse (*p* = 0.002, d = 0.88) and time main effect (*p* < 0.001, d = 1.14) in EEG responses, as both CHO and PLA mouth rinses changed the MC activation of mentally fatigued cyclists throughout the MIT. Additionally, both MF+CHO (*p* = 0.01) and MF+PLA (*p* = 0.02) induced to a greater MC activation when compared to MF trial, as MF+CHO was greater than MF at 75% of the MIT (*p* = 0.02), and MF+PLA was greater than MF at 50% (*p* = 0.02) and 75% of the MIT (*p* = 0.04). No differences were found in MC between MF+CHO and MF+PLA trials (*p* > 0.05). Figure 3 (bottom panel) depicts the results of MC activation during MIT.

Importantly, we checked for eventual solution ingestions during the mouth rinses. Comparisons revealed that spitted volumes were comparable between CHO and PLA mouth rinse sessions (*p* = 0.70), as cyclists ingested 7 mL of solution in both CHO and PLA mouth rinse conditions.

## 5. Discussion

Given the centrally mediated mental fatigue effects, we hypothesized that CHO mouth rinse could enhance PFC and MC activation and mitigate deleterious mental fatigue effects on exercise performance, as a body of evidence has shown that CHO mouth rinse may potentiate exercise performance through central pathways [17]. We observed that CHO mouth rinse was effective to mitigate the mental fatigue-reduced MIT performance, despite the comparable alterations in PFC and MC activation between CHO and PLA. These results confirmed that CHO mouth rinse might counteract mental fatigue effects on exercise performance, although challenging its role in cortical activation. Additionally, placebo mouth rinse improved the power output corresponding to ventilatory thresholds, even though mental fatigue has not impaired them.

### 5.1. Proof-of-Principle of the Mental Fatigue Effects on MIT Outcomes

Since an earlier study by Marcora et al. in 2009 [5], mental fatigue effects on endurance exercise performance have been studied in different exercise modes; however, only two included MIT protocols in mental fatigue scenarios [6,7]. Furthermore, only a small number of studies included EEG theta band analysis to verify mental fatigue effects on PFC activation, a cortical area involved in proactive, goal-directed behavior [2,3,36]. It has been suggested that mental fatigue slows down the PFC activity, probably due to the cognitive overload-induced adenosine accumulation, as adenosine accumulation may impair the PFC ability to deal with aversive feelings and attentional control [1,44]. Previous findings showed that the large increase in EEG theta band from pre to post RVIP test was associated with an increased mental fatigue sensation and impaired performance in subsequent endurance exercise [2,3]. Indeed, in the present study, we also observed that the RVIP test induced an increase in EEG theta band over the PFC and led cyclists to report a higher mental fatigue sensation. Accordingly, the mental fatigue impaired exercise performance in MIT.

The results of the effects of mental fatigue on MIT performance are controversial, as some have found that mental fatigue is deleterious to MIT performance [39,45], while others have not [7]. For example, studies have found a decrease in the total distance completed by soccer [39] and cricket players [45] during a Yo-Yo test under mental fatigue. Moreover, Zering et al. [6] observed a reduced W_PEAK_ and VO_2MAX_ in a laboratorial cycling MIT, although VT_1_ was unchanged. In contrast, Vrijkotte et al. [7] showed no negative effects of mental fatigue on a similar cycling MIT. The reason for such a controversial result involving MIT is unclear [7,39,45], perhaps methodological factors related to the exercise familiarization and experimental conditions might have influenced some results [7]. In the present study, we observed that 40 min RVIP test reduced the time to exhaustion and W_PEAK_ during MIT by around 1.9% and 2.0%, respectively, reinforcing the notion of mental fatigue-impaired performance in MIT [39,45]. Additionally, mental fatigue neither impaired VT_1_ and VT_2_, regardless of the variable used to express them, nor did change the VO_2PEAK_. While the VO_2_ results agreed with those reported elsewhere [5], thresholds results might suggest that mental fatigue was ineffective in changing these cardiopulmonary fitness variables in well-trained cyclists.

### 5.2. Effects of CHO Mouth Rinse in Mentally Fatigued Cyclists

Since CHO mouth rinse has been suggested to enhance activation in cortical structures, such as PFC areas [18,19], we hypothesized that CHO mouth rinse could enhance PFC and MC activation and MIT performance in mentally fatigued individuals.

Different from previous studies that have used manipulations to counteract the effects of mental fatigue before or during the cognitive test [2,20], in the present study, cyclists rinsed their mouth with CHO during exercise, when they were already mentally fatigued. As the use of centrally active compounds before or during the RVIP protocol [1,2] could have interfered in cognitive, perceptual, and EEG responses, we designed the CHO mouth rinses only after the RVIP test. Actually, Van Cutsem et al. [20] observed that performing caffeine-CHO mouthwash before and during the high-demanding cognitive task attenuated the mental fatigue sensation and cerebral changes, as measured by EEG, and ameliorated cognitive performance responses. Importantly, in the present study, mental fatigue effects were comparable before the exercise among MF, MF+CHO, and MF+PLA sessions.

To the best of our knowledge, only two studies have investigated the effects of centrally active compounds on exercise performance of mentally fatigued participants [2,10], and no studies have investigated if CHO mouth rinse could attenuate MIT performance reductions in mentally fatigued cyclists, despite evidence for a potential benefit of CHO mouth rinse on exercise performance [17]. Our results showed that mentally fatigued cyclists improved MIT performance by ~2.4% and ~2.3% when compared to MF and MF+PLA trials, respectively, when they rinsed their mouth with CHO. Overall, the beneficial effect of CHO mouth rinse on cycling performance has been reported as ~1.7% [17]. In the present study, the smallest worthwhile change indicated that a change of 1.3% would be needed to detect significant effects on performance.

The underlying pathways of the CHO mouth rinse effects involve the oral receptors-activated brain areas, such as PFC, orbitofrontal cortex, insula, and operculum frontal [14,18,19]. Some of them, such as PFC, composes motor planning [2,3,4] and exercise regulation areas [14,23], which are similarly affected by mental fatigue, yet in the opposite direction. Interestingly, a recent study by Franco-Alvarenga et al. [2] showed that mentally fatigued cyclists improved their performance when they ingested caffeine, although regardless of alterations in PFC activity. We also observed that CHO mouth rinse improved MIT performance of mentally fatigued cyclists regardless of alterations in PFC and MC activation, as both CHO and PLA mouth rinses similarly improved PFC and MC activation. Somehow, these results challenge the central pathways suggested for CHO mouth rinse [18]. Alternatively, they may be related to the CHO-perceived PLA design used in the present study.

We used a deceptive placebo, as participants were misinformed about the presence of CHO in both mouth rinses sessions. Therefore, they expected a potential benefit when rinsing their mouth with CHO in both sessions. Studies have suggested that participants may improve physical performance when they believe they are receiving the active substance in the placebo session [31,33]. Furthermore, the placebo perceived as the active substance has the potential to induce physiological and brain responses in the same direction of the active substance [31,46]. Thus, these results of cerebral activation may also be related to our placebo design.

It has been traditionally suggested that mental fatigue-reduced performance is associated with a higher than normal RPE, irrespective of alteration in physiological responses, such as VO_2_ and heart rate [1,2,5]. Accordingly, our results indicated similar cardiopulmonary responses, given the comparable VO_2_, VE, and HR responses among MF, MF+CHO, and MF+PLA trials. Moreover, given that RPE was comparable among these trials, but the performance was reduced with mental fatigue manipulation, one may argue that there was a higher than normal RPE in the MF trial.

Importantly, PLA but not CHO mouth rinse improved the power output at VT_1_ and VT_2_. The unaltered VT_2_ power output combined with an improved W_PEAK_ in the MF+CHO trial might suggest that mentally fatigued cyclists increased the ability to tolerate exercise acidosis-derived aversive sensation after the VT_2_ occurrence [47], thus elongating the time spent between VT_2_ and W_PEAK_. Perhaps, mentally fatigued cyclists improved their capacity to resist to intensified body responses, such as hyperventilation, muscle acidosis, cerebral oxygenation, etc., after the VT_2_, when they rinsed their mouth with CHO solution [47]. From a practical perspective, these results are relevant and suggest that centrally active compounds may improve MIT outcomes in mentally fatigued individuals, regardless of potential deleterious mental fatigue effects on these outcomes. Interestingly, we also observed that cyclists improved the power output corresponding to VT_1_ and VT_2_. These results are counterintuitive and require more investigation.

### 5.3. Methodological Aspects

In the present study, we used a double-controlled experimental design having a baseline and a placebo session as controls. However, instead of using a traditional randomized placebo-controlled clinical trial in which individuals have 50% chances of using CHO vs. 50% changes in using a placebo, we used a CHO-perceived placebo design, as recently suggested elsewhere [29]. Thus, we controlled the active substance-derived expectation effects [29,48]. Although this design has controlled expectations-induced variations in performance responses [31,49], it may have induced particular cerebral responses that may not have been present in a traditional placebo-controlled clinical trial [31], as discussed earlier.

Although we rigorously controlled the internal validity of the study (i.e., characteristic of cyclists, equipment used, environment, etc.) and provided EEG signal with reasonable quality during exercise at increasing controlled intensities, we did not analyze EEG responses from 75% of the W_PEAK_ because artifacts derived from vigorous head and trunk movements could limit the signal quality [50]. Besides, no mouth rinse was performed from this intensity, as the hyperventilation present in the last MIT intensities would have hindered the use of controlled mouth rinses (i.e., no ingestion). Therefore, the present study was unable to unravel the mental fatigue-CHO mouth rinse interplay from intensities above 75% of the W_PEAK_. However, considering the growing interest in investigating cortical measures during exercise [23,38], the present study provided an example of a rigorous, well-controlled methodology to assess EEG responses at different controlled intensities. In this regard, we used an MIT to investigate the potential effects of CHO mouth rinse on exercise performance in mentally fatigued individuals, as this exercise mode allowed us to know the effects of CHO mouth rinse by exercise intensity interaction on EEG and exercise performance responses [24,51]. Considering that CHO mouth rinse seems to have a timely effect [27] and that most studies have used CHO mouth rinses at regular intervals [16,18], we designed an MIT model to make regular mouth rinses possible (every 25% of the preliminary MIT).

## 6. Conclusions

In conclusion, the results of the present study showed that CHO mouth rinse is an effective strategy to mitigate deleterious mental fatigue effects on MIT performance, perhaps improving the ability to tolerate aversive sensations above the VT_2_. However, our results challenged the role of CHO mouth rinse on PFC and MC activation, highlighting a potential placebo effect on cerebral responses.

## Figures and Tables

**Figure 1 brainsci-10-00493-f001:**
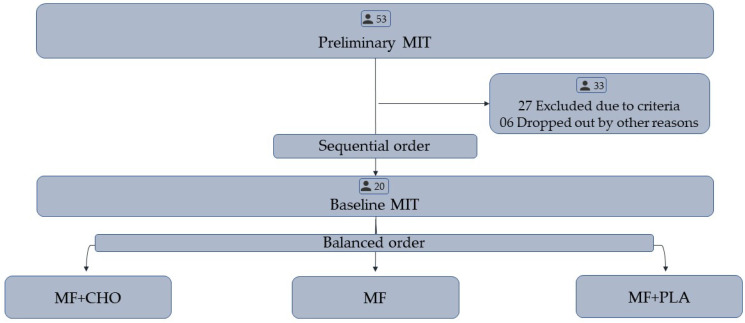
Proof-of-principle and experimental phases of the study. MIT = maximal incremental test. MF = mental fatigue session, MF+CHO = mental fatigue session rinsing carbohydrate, MF+PLA = mental fatigue session rinsing placebo perceived-as-carbohydrate.

**Figure 2 brainsci-10-00493-f002:**
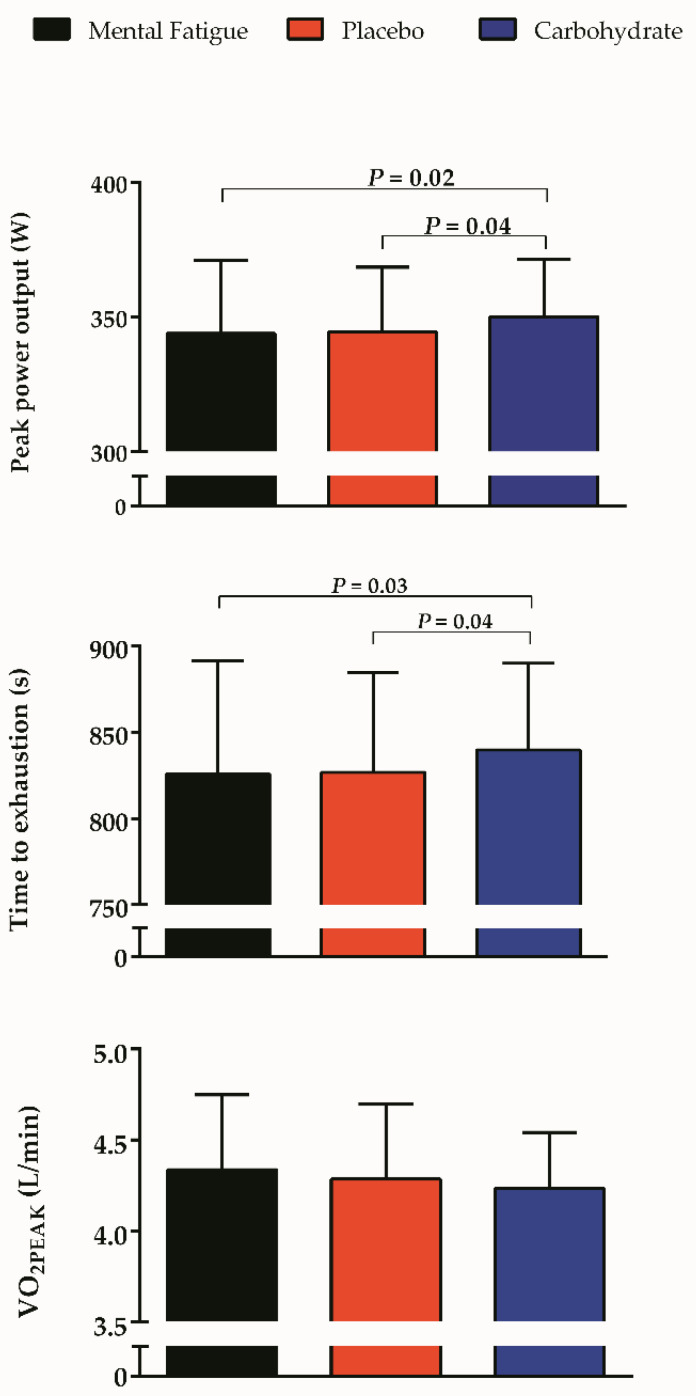
Peak power output (W_PEAK_, **upper panel**), time to exhaustion (**middle panel**), and the peak of O_2_ uptake (VO_2PEAK_, **bottom panel**) in mental fatigue, and mental fatigue combined with placebo and carbohydrate mouth rinse.

**Figure 3 brainsci-10-00493-f003:**
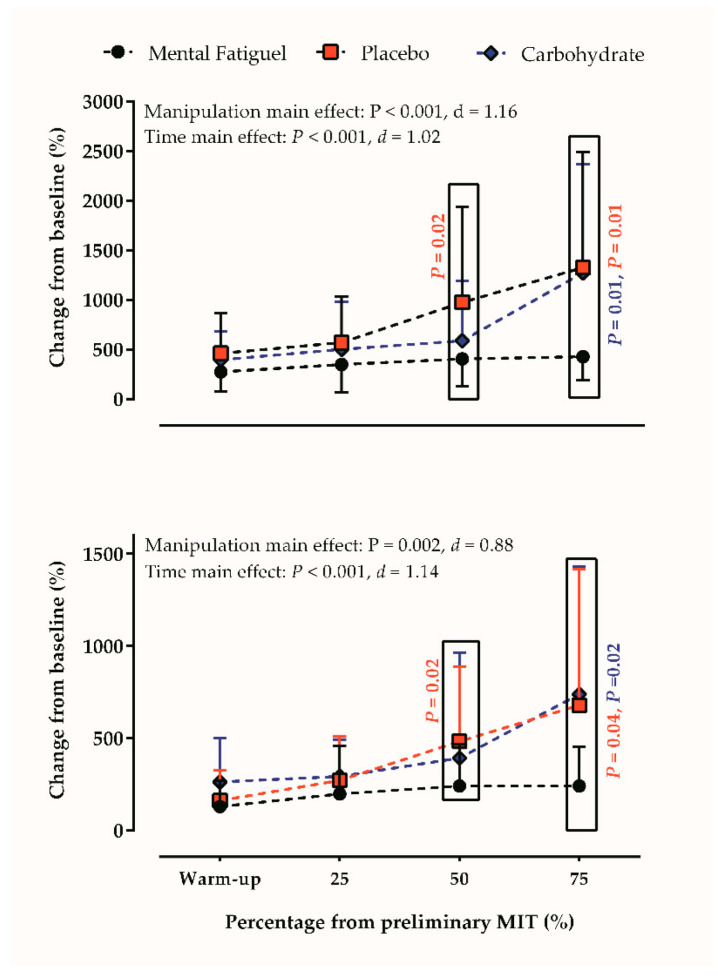
The prefrontal cortex (PFC, **upper panel**) and motor cortex (MC, **bottom panel**) activity throughout the maximal incremental test (MIT) during mental fatigue and mental fatigue combined with carbohydrate and placebo mouth rinse.

**Table 1 brainsci-10-00493-t001:** Proof-of-principle study of the mental fatigue effects on maximal incremental test outcomes.

	Baseline	Mental Fatigue
W_PEAK_	350.84 ± 24.83	343.85 ± 27.38 *
Time to Exhaustion (seconds)	841.85 ± 59.65	827.75 ± 68.62 *
VO_2PEAK_ (L/min)	4.35 ± 0.37	4.33 ± 0.41
RER_MAX_	1.16 ± 0.07	1.16 ± 0.06
RPE_MAX_	19.50 ± 0.63	19.40 ± 0.99
RPE_SLOPE_	0.89 ± 0.20	0.91 ± 0.21
VT_1_ (W)	225.10 ± 27.06	225.21 ± 25.55
VT_2_ (W)	283.22 ± 30.03	283.59 ± 29.01
VT_1_ (% W_PEAK_)	62.99 ± 7.12	64.62 ± 5.77
VT_2_ (% W_PEAK_)	80.11 ± 7.19	82.33 ± 6.79
VT_1_ (L/min)	3.04 ± 0.41	3.07 ± 0.39
VT_2_ (L/min)	3.57 ± 0.38	3.73 ± 0.39
VT_1_ (% VO_2PEAK_)	69.87 ± 7.15	70.87 ± 7.95
VT_2_ (% VO_2PEAK_)	82.21 ± 6.37	85.71 ± 4.71

* significantly different in W_PEAK_ (*p* = 0.002) and time to exhaustion (*p* = 0.002). W_PEAK_ = peak power output, VO_2PEAK_ = peak oxygen comsumption, RER = respiratory exchange ratio, RPE = ratings of perceived effort, VT = ventilatory threshold.

**Table 2 brainsci-10-00493-t002:** Maximal incremental test outcomes during mental fatigue, and mental fatigue combined with carbohydrate and placebo mouth rinses.

	Mental Fatigue	Placebo	Carbohydrate
RER_MAX_	1.16 ± 0.06	1.17 ± 0.07	1.16 ± 0.07
RPE_MAX_	19.40± 0.99	19.42 ± 1.50	19.79 ± 0.54
RPE_SLOPE_	0.91 ± 0.21	0.92 ± 0.18	0.88 ± 0.16
VT_1_ (W)	225.21 ± 25.55	242.43 ± 25.48	237.73 ± 17.52
VT_2_ (W)	283.59 ± 29.01	297.26 ± 26.63	297.92 ± 19.69
VT_1_ (% W_PEAK_)	64.62 ± 5.77	70.35 ± 5.49	66.78 ± 7.09
VT_2_ (% W_PEAK_)	82.33 ± 6.79	86.24 ± 4.32	84.40 ± 3.39
VT_1_ (L/min)	3.07 ± 0.39	3.30 ± 0.40	3.12 ± 0.40
VT_2_ (L/min)	3.73 ± 0.39	3.82 ± 0.38	3.85 ± 0.23
VT_1_ (% VO_2PEAK_)	70.87 ± 7.95	74.78 ± 8.31	71.57 ± 8.70
VT_2_ (% VO_2PEAK_)	85.71 ± 4.71	87.13 ± 5.12	90.20 ± 4.69

There was a main condition effect (mental fatigue vs. placebo) for VT expressed as %W_PEAK_.

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
