# Peer review of "Carbohydrate Mouth Rinse Mitigates Mental Fatigue Effects on Maximal Incremental Test Performance, but Not in Cortical Alterations"

_brainsci, 2020, doi:10.3390/brainsci10080493_

Round 1

Reviewer 1 Report

This study investigated the influence of a CHO and PLA mouth rinse on prefrontal cortex (PFC) activation, motor cortex (MC) activity, and maximal incremental cycling performance following a task eliciting mental fatigue. Strengths of this study include the study design, which accounted for any CHO-perceived placebo design, as well as the measures of PFC and MC activation at 25%, 50%, and 75% peak Watts.

This topic is important as it expands on prior investigations into the mechanisms explaining CHO mouth rinse efficacy and applies this ergogenic method to a common exercise, intermittent cycling. A few concerns exist, however, regarding how the conclusions are reached as well as how the conclusions are explained.

Major Concerns

  1. The title of the manuscript, as well as the bulk of the conclusions lead one to believe that CHO rinsing (as opposed to mouth rinsing with or without CHO) influenced mental fatigue. I’m not convinced this is what the data show and would appreciate the authors clarifying (both to me as well as the reader) how this conclusion is reached. The authors state that EEG responses in PFC and MC indicate mental fatigue (line 88). In other words, mental fatigue is assessed from PFC and MC data. When looking at the results, we see no evidence that CHO influenced PFC and MC data. Rather, mouth rinsing (regardless of CHO) influenced PFC and MC. Therefore, don’t the data support a title that, “Mouth rinsing mitigates mental fatigue, while CHO improves maximal incremental test outcome”? In other words, can we say with confidence that CHO influenced mental fatigue? If not, please make sure that the title, as well as the conclusions, make clear that CHO may not have influenced mental fatigue though it did appear to improve performance. Put another way, the results indicate two separate findings: 1) Mouth rinsing (regardless of the presence of CHO) improves mental fatigue, and 2) CHO-rinsing improves MIT performance.
  2. Line 272. The authors use Hopkins’s “smallest worthwhile change” to assess meaningful differences in performance outcomes. According to some evidence, however, the use of magnitude-based inferences should not be used for assessing performance outcomes (PMID 25051387) and at least one journal specializing in exercise science (MSSE) has banned the use of this statistical model in any of its published research. Please therefore exclude this from the present study or justify the use of the smallest worthwhile change in light of PMID 25051387. If excluding the model, the issue arises in explaining that a difference of 7 W between BASE and MF (and presumably a similar difference between CHO and PLA though the authors do not report these values) is meaningful. Please consider reporting data on within-subject daily variation. Is it possible that the 7 W difference was merely a matter of daily variance? If the difference in Watts was due to daily variation and not because of any CHO rinse, this then would correspond with the lack of difference observed in VO2peak.

Minor Concerns

  1. Line 46. Please explain to readers in the Introduction what a maximal incremental test is.
  2. Lines 81 – 92. These lines appear to be methodological considerations. Please consider moving them to the “Electroencephalography” section.
  3. Line 100. VO2max is indicated, yet VO2 is not defined until line 206. Please make sure that any abbreviations are defined before using. (For example, “ventilatory thresholds” is used in line 100 but no abbreviation is provided. Then the use of VT appears in the next paragraph). Additionally, though the authors indicate that the study addresses the influence of CHO mouth rinse on VO2max, they investigated VO2peak. Please be consistent with the use of VO2peak as VO2max was not measured.
  4. Line 113. What study(s) was used to predict treatment magnitude for sample size calculation?
  5. Lines 123–138. Please consider a major re-write in these lines, which are crucial to understanding the study. In lines 129 and 130 the authors use the term “discontinued MIT” before explaining what this is. Without knowing what “discontinued” means, understanding anything thereafter becomes difficult. Then again in line 130, the term “MF” is used and (forgive me if I’ve overlooked it) I could not find where this abbreviation was defined. In line 131 it looks like the authors go on to explain that “discontinued” means a 2 min discontinued step was used. But what now does that mean? Does this mean that cyclists stopped riding for 2 min, or merely continued at this workload for 2 min (as opposed to the other stages with were 1 min)? The authors go on to say that sessions 3, 4, and 5 were performed in counterbalanced order. But they then say that session 3 (and 2) allowed the researchers to examine mental fatigue “before studying” the mouth rinse. So was session 3 performed before 4 and 5, or was it sometimes performed after 4 and 5? The authors include Figures 1 and 2 to help understand the study design, but these figures creat further confusion and, if the text is clear, could be removed entirely. For example, in Figure 1, why is the “MF” visit (MF not defined in the figure legend) placed so far above the CHO+MF and PLA+MF symbols? Does this symbolize something? Again, I think Figures 1 and 2 could be removed entirely if the text were clear.
  6. Line 157. Were seat and handlebar positions recorded and kept the same between visits?
  7. Line 159. Exhaustion is defined as the “incapacity to maintain” pedal cadence. But how long designated “incapacity”? If pedal cadence fell below 80 rpm for 5 seconds was that determined as exhaustion?
  8. Line 196. How much time separated the mental fatiguing task from cycling exercise? Could that have influenced outcomes?
  9. Line 214. VO2peak was determined as the mean VO2 over the last 30 s of the test, but some people experience a decline in VO2 at the end of the test. Wouldn’t it have been more accurate to examine the largest continuous 30 second interval instead?
  10. Line 217. Please define RER and explain the RER breakpoint.
  11. Lines 229 – 233. These lines are redundant with lines 81 – 92. Please mention only once (my suggestion would be to move lines 81 – 92 to this section and remove lines 229 – 233)
  12. Lines 253. Please define RPE.
  13. Table 1. Please report RER max, which tells the reader a bit more about whether VO2max was achieved.
  14. Table 1. There are two “dots” where VT1 (% Wpeak) and Mental fatigue meet (i.e. 64..62)
  15. Line 318. This line states there was “no mouth rinse main effect throughout RVIP test” but at this point, there was no mouth rinse, correct? In other words, during the RVIP test there was no difference in protocol between the conditions (because they hadn’t yet performed any mouth rinses). Please make sure it’s clear to readers that mouth rinsing did not start until after the RVIP.
  16. Figure 3. I’m confused on why this is a figure. Traditionally, figures are left for primary outcomes. But this figure seems intuitive. This is prior to any mouth rinse, so why would RVIP results differ between conditions? (similar to above statement). Text is sufficient for these results.
  17. Lines 327. Please provide Wpeak values (as you did for BASE vs. MF trials)
  18. Line 362. Should be Figure 5 (not Figure 4)

Author Response

Reviewer Comments:

Reviewer #1:

This study investigated the influence of a CHO and PLA mouth rinse on prefrontal cortex (PFC) activation, motor cortex (MC) activity, and maximal incremental cycling performance following a task eliciting mental fatigue. Strengths of this study include the study design, which accounted for any CHO-perceived placebo design, as well as the measures of PFC and MC activation at 25%, 50%, and 75% peak Watts.

This topic is important as it expands on prior investigations into the mechanisms explaining CHO mouth rinse efficacy and applies this ergogenic method to a common exercise, intermittent cycling. A few concerns exist, however, regarding how the conclusions are reached as well as how the conclusions are explained.

Major Concerns

Thank you very much for your careful review, we appreciate it.

The title of the manuscript, as well as the bulk of the conclusions lead one to believe that CHO rinsing (as opposed to mouth rinsing with or without CHO) influenced mental fatigue. I’m not convinced this is what the data show and would appreciate the authors clarifying (both to me as well as the reader) how this conclusion is reached. The authors state that EEG responses in PFC and MC indicate mental fatigue (line 88). In other words, mental fatigue is assessed from PFC and MC data. When looking at the results, we see no evidence that CHO influenced PFC and MC data. Rather, mouth rinsing (regardless of CHO) influenced PFC and MC. Therefore, don’t the data support a title that, “Mouth rinsing mitigates mental fatigue, while CHO improves maximal incremental test outcome”? In other words, can we say with confidence that CHO influenced mental fatigue? If not, please make sure that the title, as well as the conclusions, make clear that CHO may not have influenced mental fatigue though it did appear to improve performance. Put another way, the results indicate two separate findings: 1) Mouth rinsing (regardless of the presence of CHO) improves mental fatigue, and 2) CHO-rinsing improves MIT performance.

Thank you for your thoughts. Mental fatigue state is diagnosed through the combination of one or more responses such as: a) an increased fatigue sensation; b) increased EEG theta wave in PFC; c) impaired (often, but not always) cognitive performance. These measures are normally obtained at rest and indicate how mentally fatigued the participant is. As shown by our results, all these measures indicated that participants were mentally fatigued after the cognitive test, regardless of the intervention (given that the mouth rinse was administered when participants were already exercising, from the warmup). It is important to point out that EEG theta band (3 – 7 Hz) over the PFC, at rest, is a sensitive tool to detect a mental fatigue state (as informed in the introduction). Thus, we used the EEG theta band measured over the PFC at rest (together with fatigue sensation and cognitive performance) to check if mental fatigue was present. However, analysis of EEG theta band during exercise seems to inform little about the exercise-induced alterations in cortical responses. Instead, analysis of the total EEG spectral energy (3 – 50 Hz) over PFC and MC areas may be more adequate to assess the exercise-induced alterations in areas involved with motor planning and command during exercise. Hence, the complex interplay between mouth rinse vs mental fatigue effects on cortical responses during exercise was assessed through the total EEG spectral energy (3 – 50 Hz) over PFC and MC areas. In this sense, we observed that both mouth rinses changed the exercise-induced cortical responses in mentally fatigued individuals (as depicted in our fig. 3), as activation in PFC and MC was different in both placebo and CHO mouth rinses when compared with mental fatigue. However, only the CHO mouth rinse was capable to improve the exercise performance. Therefore, based on these results we concluded that CHO mouth rinse improved exercise performance in mentally fatigued individuals, irrespective of changes in cortical responses. You have raised an important point, drawing our attention to clarify theses points. We have some amendments along the text (introduction, methods and discussion) to clarify them. Please, let us know if it is clear now.

Line 272. The authors use Hopkins’s “smallest worthwhile change” to assess meaningful differences in performance outcomes. According to some evidence, however, the use of magnitude-based inferences should not be used for assessing performance outcomes (PMID 25051387) and at least one journal specializing in exercise science (MSSE) has banned the use of this statistical model in any of its published research. Please therefore exclude this from the present study or justify the use of the smallest worthwhile change in light of PMID 25051387. If excluding the model, the issue arises in explaining that a difference of 7 W between BASE and MF (and presumably a similar difference between CHO and PLA though the authors do not report these values) is meaningful. Please consider reporting data on within-subject daily variation. Is it possible that the 7 W difference was merely a matter of daily variance? If the difference in Watts was due to daily variation and not because of any CHO rinse, this then would correspond with the lack of difference observed in VO2peak.

Thank you! We agree with you that concerns involving the magnitude-based inference (MBI) may limit its use (even though we feel that this is not a totally solved question by the scientific community, as some journals have banned the use of MBI approach, but other did not - IJSPP and the own Brain Sciences). Anyways, although we have estimated the smallest worthwhile change through an equation that is frequently combined with the MBI calculation, we did not use the MBI itself to base our conclusions. Rather, we calculated the smallest worthwhile change only to indicate the minimum size of the difference that would be scientifically sound to this scientific scenario. We provided a complementary “qualitative approach” which should be viewed together with the traditional inferential stats. Indeed, our results that a ~7 W difference (2% reduction in performance) was a worthwhile change in performance due to mental fatigue dropped within the range of mental fatigue effects on endurance performance (2% to 16% reduction), according to a systematic review (Cutsen et al., 2017 - DOI 10.1007/s40279-016-0672-0), getting close to values reported by a study (~2.4%) using a similar cycling test (Zering et al., 2017 - 10.1080/02640414.2016.1237777). Hence, while one may argue that a 7W difference in Wpeak may lack meaning in real world, this represented the minimal effect usually reported by studies investigating mental fatigue effects on endurance performance. Furthermore, the inclusion criteria used in the present study produced a less heterogeneous sample (WPEAK ≥ 325 W), perhaps reinforcing the notion that a ~7W difference is meaningful to those participants investigated in the present study. Also, a within-subject daily variation would require a repeated-measure design having the same experimental intervention, so that the presence of MF vs MF+PLA vs MF+CHO in our experimental sessions limit the use of this approach in the present study.

Minor Concerns

Line 46. Please explain to readers in the Introduction what a maximal incremental test is.

We inserted a sentence to improve this description, accordingly (please, see lines 48-50).

Lines 81 – 92. These lines appear to be methodological considerations. Please consider moving them to the “Electroencephalography” section.

Given that EEG analysis to investigate mental fatigue effects may differ from EEG analysis to investigate the exercise-induced changes itself, we think this description in the introduction is helpful to readers to understand the rationale behind our approach. In addition, the introduction is not long so we feel this may help readers to better understand the approach without taking the introduction as a hard business. In any case, we shortened this part in the introduction. Please, let us know if you are happy with this solution.

Line 100. VO2max is indicated, yet VO2 is not defined until line 206. Please make sure that any abbreviations are defined before using. (For example, “ventilatory thresholds” is used in line 100 but no abbreviation is provided. Then the use of VT appears in the next paragraph). Additionally, though the authors indicate that the study addresses the influence of CHO mouth rinse on VO2max, they investigated VO2peak. Please be consistent with the use of VO2peak as VO2max was not measured.

Sorry! Thank you for calling attention to this. VO2max was originally defined at the end of the first paragraph (which includes VO2 definition). We have revised the VO2peak acronyms thoroughly.

Line 113. What study(s) was used to predict treatment magnitude for sample size calculation?

Pottier et al. PMID: 19000099. We also performed some simulations in GPower v.3.1.9.2 software, obtaining a sample size of 18 participants. We then used the more conservative estimation.

Lines 123–138. Please consider a major re-write in these lines, which are crucial to understanding the study. In lines 129 and 130 the authors use the term “discontinued MIT” before explaining what this is. Without knowing what “discontinued” means, understanding anything thereafter becomes difficult. Then again in line 130, the term “MF” is used and (forgive me if I’ve overlooked it) I could not find where this abbreviation was defined. In line 131 it looks like the authors go on to explain that “discontinued” means a 2 min discontinued step was used. But what now does that mean? Does this mean that cyclists stopped riding for 2 min, or merely continued at this workload for 2 min (as opposed to the other stages with were 1 min)? The authors go on to say that sessions 3, 4, and 5 were performed in counterbalanced order. But they then say that session 3 (and 2) allowed the researchers to examine mental fatigue “before studying” the mouth rinse. So was session 3 performed before 4 and 5, or was it sometimes performed after 4 and 5? The authors include Figures 1 and 2 to help understand the study design, but these figures creat further confusion and, if the text is clear, could be removed entirely. For example, in Figure 1, why is the “MF” visit (MF not defined in the figure legend) placed so far above the CHO+MF and PLA+MF symbols? Does this symbolize something? Again, I think Figures 1 and 2 could be removed entirely if the text were clear.

Thank you! We have reorganized this section, making clear what means discontinued steps. We also removed fig. 2, as we agree that detail in the text make the fig unnecessary. However, we think fig. 1 (study design) may help the readers to clearly understand the design. In addition, we had used the acronym “MF” only to refer to the experimental conditions with mental fatigue. Now, MF is used consistently used to indicate mental fatigue. Regarding the order, you are right. As explained in the text, sessions 3-5 were in balanced order, thus sometimes session 3 was performed before, sometimes after sessions 4 and 5 (according the balance). To perform the proof-of-principle phase, comparisons were made between session 2 (BASE) and 3 (MF, irrespective of the order). Captions (MF, CHO+MF and PLA+MF) were included in the figure 1. Please, let us know if you are happy with this.

Line 157. Were seat and handlebar positions recorded and kept the same between visits?

Yes, it was maintained in all visits. Cyclists adjusted seat and handlebar in the preliminary session then maintained adjusts in other sessions (line 164).

Line 159. Exhaustion is defined as the “incapacity to maintain” pedal cadence. But how long designated “incapacity”? If pedal cadence fell below 80 rpm for 5 seconds was that determined as exhaustion?

Exhaustion was determined if they could no longer maintain the cadence despite 3 verbal encouragement, approximately 5 s (line 166-167).

Line 196. How much time separated the mental fatiguing task from cycling exercise? Could that have influenced outcomes?

Cyclists moved to the bike immediately after the resting EEG acquisition (3 min acquisition). Thus, the time taken from the mental fatigue test completion to the cycling warm-up was ~5 minutes, being within the time normally reported (PMID 29615923 and 30742838).

Line 214. VO2peak was determined as the mean VO2 over the last 30 s of the test, but some people experience a decline in VO2 at the end of the test. Wouldn’t it have been more accurate to examine the largest continuous 30 second interval instead?

Thanks for your comment. We used the following definition: “determination of VO2peak as average of 30s at presumed maximal effort”, as suggested elsewhere (PMID 28510504). We checked the VO2peak determination through different criteria, we noted that these result in minimal differences in absolute values. Please, let us know if you are comfortable with this.

Line 217. Please define RER and explain the RER breakpoint.

Thanks for your comment. Reported, accordingly (line 222).

Lines 229 – 233. These lines are redundant with lines 81 – 92. Please mention only once (my suggestion would be to move lines 81 – 92 to this section and remove lines 229 – 233)

Lines 253. Please define RPE.

RPE is defined in the line 143. Thank you.

Table 1. Please report RER max, which tells the reader a bit more about whether VO2max was achieved.

Ok, RERmax is now reported in both Table 1 (proof-of-principle) and Table 2 (experimental mouth rinses). In addition, we also reported RPEmax.

Table 1. There are two “dots” where VT1 (% Wpeak) and Mental fatigue meet (i.e. 64..62)

Thanks, corrected accordingly.

Line 318. This line states there was “no mouth rinse main effect throughout RVIP test” but at this point, there was no mouth rinse, correct? In other words, during the RVIP test there was no difference in protocol between the conditions (because they hadn’t yet performed any mouth rinses). Please make sure it’s clear to readers that mouth rinsing did not start until after the RVIP.

You are right, thank you! It was rewritten (lines 332-334).

Figure 3. I’m confused on why this is a figure. Traditionally, figures are left for primary outcomes. But this figure seems intuitive. This is prior to any mouth rinse, so why would RVIP results differ between conditions? (similar to above statement). Text is sufficient for these results.

Thank you. Figure 3 was removed and figures reordered.

Lines 327. Please provide Wpeak values (as you did for BASE vs. MF trials)

Values are now provided (lines 329-332).

Line 362. Should be Figure 5 (not Figure 4)

Changed, accordingly.

Reviewer 2 Report

Line 64: you mention ‘a number of studies’ but present only one reference

Line 110: explain how the subjects were recruited and add the criteria for inclusion and exclusion. You also need to add the sex of the subjects

Line 211: how have you controlled ‘rinse their mouth’ without swallowing?

Lines 371-372: you mention ‘a body of evidence’ but present only one reference

Author Response

Line 64: you mention ‘a number of studies’ but present only one reference

Thanks for your comment. The cited reference in this sentence is a systematic review (ref 17) which represent a body of literature.

Line 110: explain how the subjects were recruited and add the criteria for inclusion and exclusion. You also need to add the sex of the subjects

Thanks. It is now described (lines 113-121).

Line 211: how have you controlled ‘rinse their mouth’ without swallowing?

Thanks. It is now explained in the lines 187-189.

Lines 371-372: you mention ‘a body of evidence’ but present only one reference

As in the line 64, the cited reference in this sentence (ref 17) is a systematic review which represent a body of literature.

Round 2

Reviewer 1 Report

Thank you for addressing all comments. Good luck in the continued review process.